

# Inter-comparison of NIOSH and IMPROVE protocols for OC and EC determination: Implications for inter-protocol data conversion

**Cheng Wu[1], X. H. Hilda Huang[1], Wai Man Ng[2], Stephen M. Griffith[3], Jian Zhen Yu[1, 2, 3]**

[1] Division of Environment, Hong Kong University of Science and Technology, Clear Water Bay, Hong Kong, China

[2] Environmental Central Facility, Hong Kong University of Science and Technology, Clear Water Bay, Hong Kong, China

[3] Department of Chemistry, Hong Kong University of Science and Technology, Hong Kong, China

## Abstract

Organic Carbon (OC) and Elemental Carbon (EC) are operationally defined by analytical methods. As a result, OC and EC measurements are protocol dependent, leading to uncertainties in their quantification. In this study, more than 1300 Hong Kong samples were analyzed using both NIOSH TOT and IMPROVE TOR protocols to explore the cause of EC disagreement between the two protocols. EC discrepancy mainly (83%) arises from a difference in peak inert mode temperature, which determines the allocation of $OC4_{NSH}$, while the rest (17%) is attributed to a difference in the laser signal (transmittance vs. reflectance) applied for the charring correction. Evidence shows that the magnitude of the EC discrepancy is positively correlated with the intensity of biomass burning signal, whereby biomass burning increases the fraction of $OC4_{NSH}$, and widens the disagreement in the inter-protocol EC determination. It is also found that the EC discrepancy is positively correlated with the abundance of metal oxide in the samples. Two approaches (M1 and M2) that translate NIOSH TOT OC and EC data into IMPROVE TOR OC and EC data are proposed. M1 uses direct relationship between $EC_{NSH\_TOT}$ and $EC_{IMP\_TOR}$ for reconstruction,

$$M1: \qquad EC_{IMP\_TOR} = a \times EC_{NSH\_TOT} + b$$

while M2 deconstructs $EC_{IMP\_TOR}$ into several terms based on analysis principles, and apply regression only on the unknown terms.

$$M2: \qquad EC_{IMP\_TOR} = AEC_{NSH} + OC4_{NSH} - (a \times PC_{NSH\_TOR} + b)$$

where $AEC_{NSH}$, apparent EC by NISOH protocol, is the carbon that evolves in $He/O_2$ analysis stage, $OC4_{NSH}$ is the carbon that evolves at the fourth temperature step of the He only analysis stage, and $PC_{NSH\_TOR}$ is the pyrolyzed carbon as determined by NIOSH protocol. The implementation of M1 to all urban site data (without considering site or seasonal specificity) yields the following equation,

$$M1\ (urban\ data): \qquad EC_{IMP_{TOR}} = 2.20 \times EC_{NSH_{TOT}} - 0.05$$

While both M1 and M2 are acceptable, M2 with site-specific parameters provides the best reconstruction performance. Secondary OC (SOC) estimation using OC and EC by the two protocols is compared. An





analysis of the usability of reconstructed $EC_{IMP\_TOR}$ and $OC_{IMP\_TOR}$ suggests that the reconstructed values are not suitable for SOC estimation due to poor reconstruction of the OC/EC ratio.

## 1. Introduction

Carbonaceous aerosols are one of the major components of fine particulate matter ($PM_{2.5}$) in urbanized areas as a result of intense anthropogenic emissions. Carbonaceous aerosols consist of three
categories: organic carbon (OC), elemental carbon (EC) and carbonate carbon (CC). OC can be either primary or secondary in origin, but EC is exclusively from primary emission. CC is only abundant in
regions affected by mineral dust outflow and is negligible in other areas. OC and EC not only contribute to the overall $PM_{2.5}$ load, but these components have specific public health concerns because of their
interactions with the human body (Dou et al., 2015; Shi et al., 2015), and significantly contribute to visibility degradation (Malm et al., 1994) and climate forcing (Bond et al., 2011).

Differentiating OC and EC is still challenging due to their complex chemical structure and optical

properties. The most widely used technique to separate OC and EC is thermal optical analysis (TOA), which involves volatilizing the OC from a substrate while increasing the temperature by steps in an inert
He-only atmosphere followed by combusting EC component in an oxygenated He atmosphere. A correction for charred OC (pyrolysis carbon, PC) in the inert stage relies on continuous laser transmittance
or reflectance of the filter. However, the separation of OC and EC in TOA is operationally defined due to the lack of widely accepted reference materials for calibration. A variety of TOA protocols are used by
different research groups and sampling networks (Watson et al., 2005). Among the TOA protocols, NIOSH (Birch and Cary, 1996) and IMPROVE (Chow et al., 1993) are most widely applied, which differ in their
temperature ramping and optical correction schemes (Table S1).
Previous studies suggest that total carbon (TC), which is the sum of OC and EC, agrees very well (Chow et al., 2001) between the two protocols, but measured EC differs by a factor of 2~10, depending on
the source and aging of the samples (Chow et al., 2001; Cheng et al., 2014). The EC discrepancy between NIOSH and IMPROVE mainly arises from the temperature ramping regime and the charring correction.
The peak inert mode temperature (PIMT) in NIOSH (870 ℃) is much higher than in IMPROVE (550 ℃). Thus, NIOSH may be subject to premature EC evolution (i.e. underestimation of EC), but IMPROVE may
overestimate EC following incomplete OC evolution in the inert atmosphere (Piazzalunga et al., 2011). Since the optimal PIMT could vary between samples, a universal PIMT does not exist to avoid both of
these biases (Subramanian et al., 2006). In addition, IMPROVE uses a laser reflectance signal to perform the charring correction (TOR, thermal optical reflectance), while NIOSH adopts a laser transmittance for
charring correction (TOT, thermal optical transmittance). Correction by reflectance only accounts for charring at the filter surface (Chow et al., 2004) while the transmittance correction considers charring
throughout the filter, leading to a discrepancy in reporting pyrolysis carbon (PC).
The Pearl River Delta (PRD) is one of the most developed areas in China and home to the biggest city-clusters in the world (World Bank, 2015). Air pollution issues have arisen from the economic bloom
since the 1980s and pose a threat to public health (Tie et al., 2009). Although one of the biggest cites in the PRD, Hong Kong lacked an air quality objective regarding $PM_{2.5}$ until January 2014. To better understand
the variability of chemical compositions of $PM_{2.5}$, the Hong Kong Environmental Protection Department



of the Hong Kong Special Administration Region (HKEPD) has established a regular PM$_{2.5}$ speciation

monitoring program since 2011, including six monitoring sites, covering both suburban and urban

conditions. The samples collected in the three year period 2011-2013 were analyzed by Environmental

Central Facility at the Hong Kong University of Science & Technology. These samples have been analyzed

by both NIOSH TOT and IMPROVE TOR protocols, providing a unique opportunity to explore the OC

and EC determination dependency on analysis protocols, which is the focus of this study. This study aims

to answer the following questions: 1) What's the magnitude of the EC disagreement between the two

protocols for Hong Kong samples? 2) What are the contributing factors and how do they affect the EC

discrepancy? 3) Is it feasible to perform OC and EC data inter-protocol conversion? 4) If yes, can the

results be further used for secondary organic carbon (SOC) estimation?
## 2 Methods

### 2.1 Sample description

The 24-hour PM$_{2.5}$ samples were collected every six days from January 2011 to December 2013 at six

Air Quality Monitoring Sites (AQMS) in Hong Kong. The monitoring stations include Mong Kok (MK)

just beside a busy road, Central/Western (CW), Tsuen Wan (TW), Tung Chung (TC) and Yuen Long (YL)

at several meters above ground in urban areas in Hong Kong, and Clear Water Bay (WB) in a suburban

area, as shown in Figure S1. Partisol samplers (Rupprecht & Patachnick [now Thermo Fisher Scientific],

Model 2025, Albany, NY) equipped with a Very Sharp Cut Cyclone (VSCC, BGI, Waltham, MA, USA)

and operating at a flow rate of 16.7 L min$^{-1}$ were deployed at each AQMS. Two types of filter substrate

were used: quartz filter (Pall, 47mm 2500-QAT-UP-47, Ann Arbor, MI, USA) and Teflon filter (Whatman,

PTFE, 46.2 mm with support ring, Clifton, NJ, USA). Sample filters were retrieved within 24 hours and

stored in Petri dishes sealed with parafilm under freezing temperatures.

102   .

### 2.2 Sample analysis

Chemical analysis methods were described in detail by Huang et al. (2014), so only a brief description

is given here. Teflon filters were first used for gravimetric analysis for PM$_{2.5}$ mass concentrations using a

microbalance (Sartorius, MC-5, Göttingen, Germany) in a temperature and relative humidity controlled

room, then were used for elemental analysis (for more than 40 elements with atomic number ranging from

11 to 92) with an X-ray fluorescence (XRF) spectrometer (PANalytical, Epsilon 5, Almelo, the

Netherlands). Quartz filters were analyzed by ion chromatography (Dionex, ICS-1000, Sunnyvale, CA,

USA) and by TOA using a Sunset Laboratory Analyzer (Tigard, OR, USA). In TOA, both NIOSH and

IMPROVE protocols were employed for OC and EC quantification. Detailed temperature programs of the

two protocols are shown in Table S1 and example analysis thermographs are shown in Figure 1. The

carbon analyzer is capable of performing both laser transmittance and reflectance charring corrections, thus

both TOT and TOR results can be obtained for each protocol temperature program. As a result, four sets of

analysis data are obtained and used for investigation of OC and EC determination dependency on analysis

protocols in this study. The four sets of data are denoted as NIOSH TOT, NIOSH TOR, IMPROVE TOT

and IMPROVE TOR, with NIOSH and IMPROVE representing their respective temperature program and

TOT and TOR representing the mean of charring correction based on laser transmittance and reflectance,

respectively. It should be noted that NISOH TOT and IMPROVE TOR data represent data by the two

protocols while the other two sets of data are usually not reported in EC and OC analysis. The

concentrations of water-soluble organic carbon (WSOC) and three sugar compounds (levoglucosan,



mannosan, and galactosan) were available for 2013 WB samples from a separate project. WSOC
concentrations were measured by a TOC analyzer (Shimadzu TOC-V$_{CPH}$, Japan) (Kuang et al., 2015). The

sugars were analyzed by high-performance anion-exchange chromatography (HPAEC) with a pulsed
amperometric detection (PAD) method (Engling et al., 2006).


**2.3 Quality assurance/quality control of OCEC data**

Since OC and EC are operationally defined and lack reference materials, external calibration is only
performed for TC on a biweekly basis using sucrose solutions (Wu et al., 2012). Duplication analysis

covering 14% of the total samples was conducted for quality control purposes. TC by the two protocols
(NIOSH and IMPROVE) agree very well as evidenced by the unity regression slope (Figure S2a, slope =

0.99, $R^2$ = 0.99) and sharp frequency distribution of NIOSH TC/ IMPROVE TC ratios (Figure S2b).
Nevertheless, a small number of extreme data remain. The following criteria are used during the data

processing to screen out the suspicious data: 0.1<OC/EC<40; 0.5<TC$_{NSH}$/TC$_{IMP}$<2.
# 3 Results and discussion

**3.1 Ambient PM$_{2.5}$ OC and EC concentrations**

The three-year distribution of OC and EC concentrations are shown in Figure 2, where a clear spatial
gradient can be seen from the roadside site to the urban sites and suburban site. OC and EC levels at the

MK roadside site are a factor of two higher for both protocols compared to the urban sites. Annual average
concentrations and standard deviations for the five sites are listed in Table 1. Compared to samples

collected at the MK and TW sites in 2000 Nov-2001 Dec (Chow et al., 2002), both OC and EC three-year
annual average concentrations observed in this study are lower by a factor of 1.4 to 2.3. At the TW site,

TOR OC decreased from 8.69 μg m$^{-3}$ to 4.94±3.14 μgm$^{-3}$ and TOR EC decreased from 5.37 μg m$^{-3}$ to
3.97±1.84 μg m$^{-3}$. The reduction is more pronounced at the MK roadside site where TOR OC decreased

from 16.64 μg m$^{-3}$ to 7.33±3.28 μg m$^{-3}$ and TOR EC decreased from 20.29 μg m$^{-3}$ to 9.03±2.27 μg m$^{-3}$
(Chow et al., 2002).


**3.2 NIOSH and IMPROVE comparison for OC and EC determination**

The data discussed in this section use the unit of μg cm$^{-2}$, because the inter-protocol comparison focus
in the analytical aspect of OC/EC analysis that has more association with filter loading rather than air

concentration. As mentioned earlier, the difference in the peak inert mode temperature for NIOSH (870 ℃)
and IMPROVE (550 ℃) is an important distinguishing factor between the two protocols. The carbon

fraction evolved between 550 ℃ and 870 ℃ is classified as OC4 in the NIOSH protocol, while in
IMPROVE this fraction is considered part of apparent EC (AEC), which is the sum of all the EC fractions

before correcting for charred OC. Chow et al. (2001) found NIOSH OC4 can explain most of the EC
difference in US samples between the two protocols, and the relationship has since been further defined in

our PRD study and is shown in Equation 1 where IMPROVE AEC is equal to the sum of NIOSH OC4 and
AEC (Wu et al., 2012).

$$AEC_{IMP} = AEC_{NSH} + OC4_{NSH} \qquad (1)$$

HK samples from the current study also confirm this relationship as shown in Figure 3a (Slope =0.99). The

reported IMPROVE TOR EC is the sum of carbon fractions evolved in the He/O$_2$ stage minus pyrolysis
carbon (PC) as measured by laser reflectance.

$$EC_{IMP\_TOR} = AEC_{IMP} - PC_{IMP\_TOR} \qquad (2)$$



Combining equations 1 and 2, the IMPROVE TOR EC can be written as

$$EC_{IMP\_TOR} = AEC_{NSH} + OC4_{NSH} - PC_{IMP\_TOR} \qquad (3)$$

On the other hand, the reported NIOSH TOT EC is sum of carbon fractions evolved in $He/O_2$ stage minus pyrolysis carbon by laser transmittance.

$$EC_{NSH\_TOT} = AEC_{NSH} - PC_{NSH\_TOT} \qquad (4)$$

As shown in Figure 3b, the linear regression slope (2.05) of the scatter plot represents the average discrepancy between $EC_{IMP\_TOR}$ (y axis) and $EC_{NSH\_TOT}$ (x axis). As embodied in equations 3 and 4, the EC discrepancy can be attributed to two factors, $OC4_{NSH}$ (thermal effect) and the difference in PC (laser effect). By adding $OC4_{NSH}$ to the x axis (Figure 3c), the effect of $OC4_{NSH}$ between y ($EC_{IMP\_TOR}$) and x ($OC4_{NSH}+EC_{NSH\_TOT}$) is minimized as embodied in equations 3 and 5, where the slope (1.18) primarily represents the laser effect caused by the PC difference ($PC_{IMP\_TOR}$ versus $PC_{NSH\_TOT}$).

$$EC_{NSH\_TOT} + OC4_{NSH} = AEC_{NSH} + OC4_{NSH} - PC_{NSH\_TOT} \qquad (5)$$

The difference between the slopes in Figure 3b (slope=2.05) and Figure 3c (slope=1.18) indicates the contribution of the thermal effect to the EC discrepancy. By examining the relative differences from unity in the two slopes (i.e. 0.18/1.05), it is estimated that 82.86% of the EC difference by the two protocols in HK samples is attributed to the thermal effect ($OC4_{NSH}$), and the rest (17.14%) is due to the PC monitoring, arising from different laser signals used for the charring correction (transmittance or reflectance). The reduced $R^2$ in Figure 3b and Figure 3c comparing to Figure 3a suggest scattering of data points is due to the laser effect (PC). The relative contribution of the two factors in the HK samples exhibits a seasonal dependency as shown in Figure S3. In summer and fall, the laser effect accounts for ~12% of the EC discrepancy, while in winter and spring, the laser effect contribution is 35%. This is in part dictated by a lower proportion of $OC4_{NSH}$ fraction in these two seasons as shown in Figure S4, leading to an attenuated thermal effect.

It is also found that the laser effect described above exhibits a dependency on the temperature ramping step. However, PC cannot be compared directly between the two protocols because they evolve under different temperature regimes, thus the PC difference of using the TOR or TOT signal within the protocols are compared as shown in Figure 4. It is found that the ratio of $EC_{IMP\_TOR}$ /$EC_{NSH\_TOT}$ shows a dependency on $PC_{NSH\_TOT}$/$PC_{NSH\_TOR}$ ($R^2$=0.12~0.41), and the degree of correlation varies by season (Figure 4a & 4b). This result agrees well with the higher laser effect contribution during spring and winter shown in Figure S3 and discussed above. In contrast, $EC_{IMP\_TOR}$ /$EC_{NSH\_TOT}$ is insensitive to $PC_{IMP\_TOT}$/$PC_{IMP\_TOR}$ ($R^2$=0) as shown in Figure 4c. This selective dependency suggests the PC difference introduced by TOT/TOR is more pronounced on the $OC4_{NSH}$ fraction as $OC4_{NSH}$ is the only difference between potential sources of $PC_{IMP}$ and $PC_{NSH}$. Thus, the laser effect contribution to EC dependency is sensitive to the degree of charring formed during the $OC4_{NSH}$ stage.

Other potential factors affecting EC discrepancy were also examined. Cheng et al. (2011a) found in Beijing samples that biomass burning can influence the EC discrepancy. Here we use a normalized abundance of $K^+$ as an indicator to examine the impact of biomass burning on the EC discrepancy. Figure S5a is the same as Figure 3b but color coded with the $K^+$/$EC_{NSH}$ ratio to reflect the influence from biomass burning, and reveals a pattern associated with the $EC_{IMP\_TOR}$ to $EC_{NSH\_TOT}$ ratio. To verify this relationship,



regressions on the lowest and highest 10% of K⁺/EC_NSH ratios are shown in Figures S6b and S6c
respectively. The data from the highest 10% of $K^+/EC_{NSH}$ ratios has a significantly higher regression slope

(slope = 3.19, Figure S5c) than the data from the lowest 10% of $K^+/EC_{NSH}$ ratios (slope=1.48, Figure S5b),
implying the EC discrepancy depends on the $K^+/EC_{NSH}$ ratio. To further distinguish whether the $K^+/EC_{NSH}$

effect is associated with $OC4_{NSH}$ (thermal effect) or the difference in PC (laser effect), $OC4_{NSH}$ is added to
the x-axis as shown in Figure S5d~f. By adding $OC4_{NSH}$ to the x-axis, the discrepancy between y and x is

can be attributed to the laser effect alone. The slopes of samples from the highest 10% of $K^+/EC_{NSH}$ ratios
(1.20, Figure S5e) and samples from the lowest 10% of $K^+/EC_{NSH}$ ratios (1.27, Figure S5f) are very close

to the slope using all samples (1.23, Figure S5d), implying that the laser effect is not sensitive to the
$K^+/EC_{NSH}$ ratio. Consequently, the EC discrepancy dependence on the $K^+/EC_{NSH}$ ratio is very likely

associated with $OC4_{NSH}$ (thermal effect). This relationship is also verified by the ratio-ratio plot shown in
Figure 5 since the intercept in Figure S5 is relative small and their slopes can be represented by ratios. As

shown in Figure 5a, when the $K^+/EC_{NSH\_TOT}$ ratio goes up, a larger EC discrepancy is observed. While
adding $OC4_{NSH}$ to the y axis (offseting the contribution from $OC4_{NSH}$) as shown in Figure 5b, this

relationship no longer holds. $OC4_{NSH}$ fraction exhibit dependency on $K^+/EC_{NSH}$ ratio as illustrated in
Figure S6. The $OC4_{NSH}/TC$ ratio is introduced to represent the relative abundance of $OC4_{NSH}$ in samples.

Independent t test (Table S2) is performed and found the average $OC4_{NSH}/TC$ ratio of samples from the
highest 10% of $K^+/EC_{NSH}$ ratios (0.27, Figure S6c) is significantly higher (p<0.001) than the average

$OC4_{NSH}/TC$ ratio of samples from the lowest 10% of $K^+/EC_{NSH}$ ratios, revealing that the $OC4_{NSH}$ fraction
and $K^+/EC_{NSH}$ ratio are positively correlated. As discussed before, $OC4_{NSH}$ fraction can affect EC

discrepancy, which is the reason that biomass burning can influence the EC discrepancy.
A suite of laboratory studies have revealed the presence of metal oxides in aerosol samples can alter
the EC/OC ratio, by either lowing the EC oxidation temperature or enhancing OC charring (Murphy et al.,

1981; Wang et al., 2010; Bladt et al., 2014). As a result, the distribution of carbon fractions is impacted
during the analysis, affecting the inter-protocol EC discrepancy. As shown in Figure 6a and Figure S7, the

EC discrepancy positively correlates with normalized Fe abundance ($Fe/EC_{NSH\_TOT}$ ratio), suggesting that a
higher fraction of metal oxide can increase the EC divergence across the two protocols. If $OC4_{NSH}$ is added

to cancel out the discrepancy contribution from the thermal effect (Figure 6b and Figure S7), the
discrepancy due to laser effect alone shows no dependency on Fe abundance. Similar dependency is also

found in another metal oxide like Al as shown in Figure S8. These results imply that metal oxide induced
EC divergence is mainly associated with the $OC4_{NSH}$ fraction.


**3.3 Comparison of IMPROVE TOR EC reconstruction approaches for Hong Kong samples**

**3.3.1 Description of two reconstruction methods**

It is of great interest to determine the best estimation for $EC_{IMP\_TOR}$ when only NIOSH TOT analysis

is available. This study provides an opportunity to examine different empirical reconstruction approaches
for $EC_{IMP\_TOR}$ using the $EC_{NSH\_TOT}$ data. In total, four approaches are investigated, two of them are

discussed below and the other two are discussed in the SI. The first method is direct regression (M1),
which applies the relationship obtained from Figure S9 to reconstruct $EC_{IMP\_TOR}$.

$$\textbf{\textit{M}1:} \qquad EC_{IMP\_TOR} = a \times EC_{NSH\_TOT} + b \qquad (6)$$

Then, reconstructed $OC_{IMP\_TOR}$ can be obtained by subtracting reconstructed $EC_{IMP\_TOR}$ from $TC_{NSH}$,

$$Reconstructed\ OC_{IMP\_TOR} = TC_{NSH} - Reconstructed\ EC_{IMP\_TOR} \qquad (7)$$





Further reconstruction methods may deconstruct $EC_{IMP\_TOR}$ into several terms based on analysis principles, and apply regression only on the unknown terms. Since only a partial regression is involved, theoretically, this approach can provide more accurate reconstruction results. Relationships found in the last section can also be used to refine the reconstruction.

The second approach (M2) employs partial regression. In equation 3, $PC_{IMP\_TOR}$ is the only unknown term on the RHS. As shown in Figure S10, $PC_{IMP\_TOR}$ is well correlated with $PC_{NSH\_TOR}$, which is known from a NIOSH analysis. Therefore, equation 3 can be approximated as

$$\textbf{\textit{M2}}: \qquad EC_{IMP\_TOR} = AEC_{NSH} + OC4_{NSH} - (a \times PC_{NSH\_TOR} + b) \qquad (8)$$

Extra reconstruction methods (M2-1 and M3) are discussed in the SI. In brief, M2-1 is a variant of M2, which use multiple linear regression (MLR) to compute $PC_{IMP\_TOR}$ in equation 3, thus it require more compositional information as inputs (elements and water soluble ions). M3 is based on the linear relationship between ($PC_{NSH\_TOT}$-$PC_{NSH\_TOR}$) and ($PC_{NSH\_TOT}$-$PC_{IMP\_TOR}$) for reconstruction.

**3.3.2 Reconstruction 2013 OC and EC using parameters from 2011-2012 data**

In this section, blind tests are performed to compare the performance of the two reconstruction methods (M1 and M2). Data from 2011-2012 are used to obtain the necessary parameters (a and b) for M1 and M2 as shown in Eq.6 and Eq. 8 respectively. Since these parameters may vary temporally and spatially, two scenarios are considered for parameterization: scenario 1, seasonal specific parameters for each season with samples from all sites; scenario 2, site-specific parameters for all samples from a site. Detailed parameters are summarized in Table 2. These parameters are then applied on NIOSH data in 2013, and reconstructed $EC_{IMP\_TOR}$ and $OC_{IMP\_TOR}$ concentrations are calculated and compared with measured 2013 $EC_{IMP\_TOR}$ and $OC_{IMP\_TOR}$ to evaluate the performance of OC and EC reconstruction by the three scenarios. Since two scenarios are considered in each reconstruction method, there are four combinations of reconstruction results for M1 and M2.

Reconstructed EC by M1 is compared with measured EC in Figure 7a and 7b. The $R^2$ of season-specific (Figure 7a) and site-specific reconstruction (Figure 7b) are comparable with each other. Reconstructed EC are also compared with measured EC using histograms as shown in Figure S15. Mean concentration by site-specific reconstruction agrees better than the season-specific reconstruction. The frequency distribution of the relative difference of reconstructed vs. measured EC exhibits similar distribution between the season and site-specific reconstructions (Figure S16). OC reconstruction by M1 is shown in Figure 8a and 8b, revealing reconstruction by site-specific parameters can increase the $R^2$, with a tradeoff of higher average bias (slope=1.14). The seasonal or site-specific parameters yield similar reconstructed OC distributions as shown in Figure S17 and S18. The reconstructed OC/EC ratios by M1 are overestimated by a factor of two as shown by the slopes in Figure 9. The reconstructed OC/EC distribution is significantly broader than the measured OC/EC ratios as shown in Figure S19 and S20. This is an expected result of bias of constructed OC and EC are of opposite sign, leading to amplified bias in its ratio quantity.

Results of $EC_{IMP\_TOR}$ reconstruction by M2 are shown in Figure 7c and 7d. Slopes by M2 are the closest to unity, implying that M2 can provide better accuracy than M1. M2 reconstruction by site exhibits the highest $R^2$ among all reconstruction scenarios. The superior performance of M2 by site-specific





parameter is also evidenced by the sharpened distribution peak around zero for the relative difference of measured and reconstructed EC (Figure S16d). OC reconstruction by M2 using site-specific parameters
(Figure 8b) yields a higher $R^2$ than the season-specific scenario (Figure 8c). The OC relative difference distribution is sharpest in the site-specific parameters scenario as shown in Figure S18d. The reconstructed
OC/EC ratios by M2 are underestimated from 22% to 72% as shown in Figure 9 with a low $R^2$ ranging from 0.3 to 0.46. The OC/EC bias is also evidenced by significantly different histograms between
reconstructed OC/EC (a sharper peak) and measured OC/EC (Figure S19c and S19d).
From the comparisons shown above, it is obvious that M2 site-specific parameters scenario can provide the best performance in OC and EC reconstruction evidenced by regression slopes closest to unity
and the sharpest frequency distribution histograms of OC or EC differences between reconstructed and measured values. However, the OC/EC ratio is not well reproduced by the two methods, which may be
either overestimated or underestimated.

To investigate the stability of various parameters used in the three reconstruction scenarios, we also

calculate reconstruction parameters for individual years from 2011 to 2013 as well as for entire three year's dataset as listed in Table 2. The reconstruction parameters are of similar values between years, implying
these methods are robust for future reconstruction applications. The implementation of M1 to all urban site data (without site or seasonal specificity) yields the following equation and this equation is recommended
for urban site data conversion.

$$M1\ (urban\ data): \qquad EC_{IMP_{TOR}} = 2.20 \times EC_{NSH_{TOT}} - 0.05 \qquad (9)$$

For heavily trafficked roadside environment, the recommended slope and intercept are 0.99 and 3.39, respectively. For suburban environments with light EC loadings, the recommended values are 2.63 for
slope and -0.05 for intercept.

The M2 site-specific parameters exhibit weaker site dependence than the M1 method, making it more

suitable for expanding its application in other regions. As a result, M2 site-specific parameters obtained from the 3-year dataset are recommended for future reconstruction applications in Hong Kong (Table 2).
The equation for urban environments is shown below:

$$M2\ (urban\ data): \qquad EC_{IMP_{TOR}} = AEC_{NSH} + OC4_{NSH} - (2.11 \times PC_{NSH_{TOR}} - 0.03) \qquad (10)$$

We note that the $AEC_{NSH}$, $OC4_{NSH}$ and $PC_{NSH\_TOR}$ inputs required in M2 are not always available for data users, as they are typically not reported by analysis laboratories.

Monthly variations of measured and reconstructed IMPROVE TOR EC and OC are shown in Figure

S21, clearly showing the reconstructed OC and EC data can reproduce the monthly trend quite well as compared with the measured data. This demonstrates that the reconstruction equations can provide a mean
to establish temporal trends for ECOC data produced using different analysis protocols.
**3.4 Implications for secondary OC (SOC) estimation**

The EC tracer method is a widely used approach for SOC estimation since it only requires measured OC

and EC as input:

$$SOC = OC_{total} - \left(\frac{OC}{EC}\right)_{pri} \times EC - OC_{non-comb} \qquad (11)$$

where (OC/EC)$_{pri}$ is the OC/EC ratio in freshly emitted combustion aerosols, OC$_{total}$ and EC are from the



measurements, and $OC_{non-comb}$ is the OC fraction from non-combustion sources (i.e. biogenic emissions).
Since the $OC_{non-comb}$ is usually small, it is considered as zero to simplify the calculation in our study. The key to the EC tracer method is to estimate a proper $(OC/EC)_{pri}$. Our previous study proved that the Minimum R Squared method (MRS) is more accurate than the conventional subset percentile or minimum OC/EC ratio approaches(Wu and Yu, 2016). Therefore, MRS is employed for SOC calculation in this study. In this section, two aspects are discussed regarding SOC estimation: 1) Variability of OC and EC by different protocols and the impacts on SOC estimation. 2) The usability of reconstructed $EC_{IMP\_TOR}$ and $OC_{IMP\_TOR}$ for SOC estimation.

Since the proportion of different primary emission sources are expected to vary by season, $(OC/EC)_{pri}$ is calculated by MRS for each season (Table S3) using all three years of data (2011-2013). As shown in Figure 10, SOC by NIOSH TOT (mean concentration: 4.70 μg m$^{-3}$) is higher than by IMPROVE TOR protocol (mean concentration: 2.66 μg m$^{-3}$). On average, $SOC_{NSH\_TOT}$ is 1.67 times higher than $SOC_{IMP\_TOR}$ as suggested by the regression slope in Figure 10c. Although the absolute SOC concentrations by the two protocols are quite divergent, the $R^2$ (0.61) suggests that the two SOCs are moderately correlated. Water Soluble Organic Carbon (WSOC) has been recognized as a good indicator of SOC formation (Sullivan et al., 2004), but WSOC contribution from primary emission is not negligible (Graham et al., 2002). Instead of using WSOC directly, we use secondary WSOC (SWSOC) as an indicator to verify the SOC results. SWSOC can be calculated from the following equation

$$SWSOC = WSOC - Sugars \times \left(\frac{WSOC}{Sugars}\right)_{pri} \tag{12}$$

In Eq.12, sugars, which includes levoglucosan, mannosan and galactosan are used as a tracer to derive SWSOC based on the primary ratio (5.28, Figure S22) obtained from a biomass burning source profile measured in the PRD region (Lin et al., 2010). The relationship between SWSOC and SOC is examined in Figure S23. SWSOC accounts for 61% of $SOC_{NSH\_TOT}$, which is comparable with the WSOC/ $SOC_{NSH\_TOT}$ ratio observed in Beijing (50%~70%) by Cheng et al. (2011b). The SWSOC-to-$SOC_{IMP\_TOR}$ regression slope is close to unity (0.92), implying that SOC by IMPROVE TOR could be underestimated. SOC by Both $SOC_{NSH\_TOT}$ and $SOC_{IMP\_TOR}$ are well correlated with SWSOC, confirming the significant contribution of WSOC to SOC in this region. $SOC_{NSH\_TOT}$ exhibits a higher correlation ($R^2$=0.92) with WSOC than $SOC_{IMP\_TOR}$ ($R^2$=0.86), which is in good agreement with the study in Beijing (Cheng et al., 2011b), suggesting that NIOSH TOT might be more reasonable for SOC estimation.

The usability of reconstructed $EC_{IMP\_TOR}$ and $OC_{IMP\_TOR}$ for SOC estimation are investigated. To account for the temporal variations of $(OC/EC)_{pri}$, seasonal $(OC/EC)_{pri}$ are calculated using OC and EC reconstructed by M1, and M2 (Table S3). These $(OC/EC)_{pri}$ values are then subject to SOC estimation following Equation 10. It is very clear that the frequency distribution of reconstructed SOCs deviate from the SOC derived from measured OC and EC (Figure 11). The SOC by M1 is higher than original SOC, not only evidenced by average concentrations (3.53 μg m$^{-3}$ vs. 2.66μg m$^{-3}$), but also confirmed by the regression slope (1.35). On the other hand, SOC by M2 is underestimated by 30~40%. The moderate $R^2$ (Figure 11d) also suggests the SOC by reconstructed $EC_{IMP\_TOR}$ and $OC_{IMP\_TOR}$ are poorly correlated with SOC by measured $EC_{IMP\_TOR}$ and $OC_{IMP\_TOR}$. The significant bias and moderate correlations suggest that reconstructed $EC_{IMP\_TOR}$ and $OC_{IMP\_TOR}$ are not suitable for SOC estimation.

## 4 Conclusions





In this study, we use a large dataset that has good temporal (three years) and spatial coverage (roadside, urban, rural) in Hong Kong to investigate the OC and EC determination discrepancy between
NIOSH TOT and IMPROVE TOR protocols. NIOSH TOT reported lower EC (higher OC) than IMPROVE TOR. The divergence between the two protocols is attributed to two effects: thermal effect and
laser correction effect. The thermal effect is due to the higher PIMT in NIOSH (870 °C) than IMPROVE (550 °C) and the allocation of the $OC4_{NSH}$ fraction. The laser correction effect is a result of different laser
signals used by the two protocols (laser transmittance by NIOSH vs. laser reflectance by IMPROVE).
The equivalence between $AEC_{IMP}$ and sum of $OC4_{NSH}$ and $AEC_{NSH}$ is confirmed in the current study, and by offsetting the discrepancy from the thermal effect ($OC4_{NSH}$), the contribution from laser correction
can be quantified. It is found that on average the thermal effect accounted for 83% of the EC disagreement while 17% is attributed to the laser effect. The contribution of the two effects exhibit a clear seasonal
dependency, with a more pronounced laser effect in spring and winter (~35%).
The intensity of biomass burning influence can affect EC divergence between the two protocols. Samples influenced by biomass burning (evidenced by higher $K^+/EC_{NSH}$ ratio) come with higher $OC4_{NSH}$
abundance (higher $OC4_{NSH}/TC$ ratio), leading to larger EC divergence between the two protocols. Abundance of metal oxide in samples can also affect EC discrepancy, with a larger EC difference observed
when a higher fraction of metal oxide is present in the ambient samples.
Two IMPROVE TOR EC reconstruction approaches (M1 and M2) are proposed. For each approach, three parameterization scenarios are considered, including single parameter, season-specific parameters
and site-specific parameters. The single parameter implementation of M1 to all urban sites (without considering site or seasonal specificity) yield the following equation,
**M1 (urban data):**          $$EC_{IMP_{TOR}} = 2.20 \times EC_{NSH_{TOT}} - 0.05$$

Considering site-specificity yields slightly better reconstruction performance, with the site-specific slope

value varying from 2.16 to 2.33 for the urban sites. The suburban site produces a higher slope value (2.63) while the roadside (MK) data produces a noticeably lower slope value (0.99). Hence, roadside samples (i.e.,
typically significant EC loadings) need to be processed separately and applied its own site-specific parameters for reconstruction when using M1 equation. The Comparisons show that the M2 with
site-specific parameters provides the best reconstruction results and the regression parameters are given in Table 2.

SOC estimation using OC and EC by the two protocols is compared. Based on the SWSOC to SOC
ratio and correlation coefficients, it is found that SOC concentrations derived from NIOSH TOT are likely more reasonable than IMPROVE TOR. The usability of reconstructed $EC_{IMP\_TOR}$ and $OC_{IMP\_TOR}$ for SOC
estimation proves to not be suitable due to the poor reconstruction of the OC/EC ratio.

## Acknowledgments


This project is partially supported by Hong Kong Environment Protection Department (AS 10-231,

11-03973, and 12-04384). We thank HKEPD for making available the data for this work. We are indebted to Dr. Peter Louie for his relentless efforts in pushing for the best possible $PM_{2.5}$ speciation measurements





in Hong Kong.

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



**Table 1**. Ambient concentrations (µg m⁻³) of OC and EC by IMPROVE TOR and NIOSH TOT protocols


| | | (Chow et al., 2002) | (Chow et al., 2006) | (Chow et al., 2010) | Current study | | | |
|---|---|---|---|---|---|---|---|---|
| | | 2001* | 2005 | 2009 | Overall | 2011 | 2012 | 2013 |
| | | | | | (Mean ± one standard deviations) | | | |
| MK | TOR OC | 16.64 | 11.17 | 6.26 | 7.33±3.28 | 8.09±3.67 | 6.94±2.55 | 6.92±3.36 |
| | TOR EC | 20.29 | 14.11 | 10.66 | 9.03±2.27 | 8.48±2.08 | 9.21±2.74 | 9.42±1.89 |
| | TOT OC | | | | 10.72±4.3 | 11.36±4.26 | 10.24±3.94 | 10.51±4.63 |
| | TOT EC | | | | 5.24±1.59 | 4.86±1.47 | 5.53±1.42 | 5.35±1.78 |
| TW | TOR OC | 8.69 | 6.93 | 4.38 | 4.94±3.14 | 5.44±3.35 | 4.5±2.4 | 4.86±3.47 |
| | TOR EC | 5.37 | 6.25 | 3.76 | 3.97±1.84 | 4.24±1.81 | 3.62±1.99 | 4.01±1.71 |
| | TOT OC | | | | 6.77±4.01 | 7.37±4.05 | 6.1±3.33 | 6.79±4.46 |
| | TOT EC | | | | 1.88±0.9 | 1.95±0.93 | 1.76±0.91 | 1.91±0.87 |
| YL | TOR OC | | 7.23 | 4.83 | 5.16±3.63 | 5.62±3.56 | 4.77±3.02 | 4.92±4.05 |
| | TOR EC | | 6.19 | 3.48 | 4.08±2.1 | 4.56±2.48 | 3.69±1.8 | 3.92±1.87 |
| | TOT OC | | | | 7.12±4.62 | 7.92±4.69 | 6.33±3.94 | 6.88±4.92 |
| | TOT EC | | | | 1.88±0.98 | 1.89±0.9 | 1.79±0.91 | 1.95±1.12 |
| CW | TOR OC | | | | 4.48±2.98 | 4.92±2.89 | 4.12±2.64 | 4.37±3.33 |
| | TOR EC | | | | 3.48±1.79 | 3.71±1.75 | 3.24±1.94 | 3.48±1.69 |
| | TOT OC | | | | 6.12±3.64 | 6.55±3.55 | 5.55±3.27 | 6.2±4.02 |
| | TOT EC | | | | 1.57±0.93 | 1.63±0.82 | 1.54±1.03 | 1.54±0.95 |
| TC | TOR OC | | | | 4.53±3.63 | 5.13±3.69 | 4.17±2.68 | 4.27±4.23 |
| | TOR EC | | | | 3.38±2.08 | 3.65±2.3 | 3.1±1.71 | 3.37±2.14 |
| | TOT OC | | | | 6.15±4.63 | 6.88±4.74 | 5.48±3.37 | 6.03±5.39 |
| | TOT EC | | | | 1.51±0.89 | 1.53±0.91 | 1.55±0.87 | 1.46±0.91 |
| WB | TOR OC | | | | 3.46±2.65 | 3.91±2.62 | 3.07±2 | 3.37±3.13 |
| | TOR EC | | | | 2.08±1.37 | 2.43±1.42 | 1.81±1.2 | 1.96±1.39 |
| | TOT OC | | | | 4.55±3.36 | 5.07±3.33 | 3.91±2.53 | 4.62±3.93 |
| | TOT EC | | | | 0.75±0.52 | 0.86±0.5 | 0.72±0.44 | 0.67±0.58 |

* 2000 Nov -2001 Oct





**Table 2**. **Regression Parameters for OC and EC reconstruction equations**.


| | Approach[a] | | 2011-2013[b] | | 2011-2012[c] | | 2011[d] | | 2012[d] | | 2013[d] | |
|---|---|---|---|---|---|---|---|---|---|---|---|---|
| | | | a | b | a | b | a | b | A | b | a | b |
| **M1** | by season | Spring | 2.07 | 0.11 | 2.12 | -0.03 | 2.32 | 0.01 | 1.82 | 0.03 | 1.63 | 0.74 |
| | | Summer | 1.77 | 0.19 | 1.86 | 0.14 | 1.97 | 0.13 | 1.72 | 0.16 | 1.63 | 0.24 |
| | | Fall | 2.17 | 0.33 | 2.17 | 0.21 | 2.10 | 0.08 | 2.20 | 0.44 | 1.59 | 1.37 |
| | | Winter | 2.12 | 0.25 | 2.19 | 0.16 | 2.08 | 0.58 | 1.95 | 0.14 | 1.99 | 0.39 |
| | by site Roadside | MK | 0.99 | 3.39 | 1.23 | 2.20 | 0.90 | 3.86 | 1.99 | -1.81 | 0.73 | 4.87 |
| | Urban | TW | 2.16 | -0.06 | 2.35 | -0.31 | 2.39 | -0.23 | 2.15 | -0.23 | 1.75 | 0.54 |
| | | YL | 2.33 | -0.34 | 2.72 | -0.83 | 2.93 | -1.02 | 2.30 | -0.42 | 1.65 | 0.55 |
| | | CW | 2.09 | 0.13 | 2.23 | -0.02 | 2.33 | -0.11 | 2.11 | 0.07 | 1.8 | 0.44 |
| | | TC | 2.24 | -0.07 | 2.24 | -0.09 | 2.39 | -0.11 | 2.01 | -0.03 | 2.20 | 0.02 |
| | | Urban sites combined | 2.20 | -0.05 | 2.26 | -0.12 | 2.37 | -0.14 | 2.07 | -0.06 | 2.00 | 0.16 |
| | Suburban | WB | 2.63 | 0.05 | 2.65 | -0.02 | 2.69 | 0.01 | 2.55 | -0.03 | 2.74 | 0.10 |
| **M2** | by season | Spring | 1.92 | 0.04 | 2.19 | 0.00 | 2.26 | 0.03 | 0.94 | 0.13 | 0.98 | 0.34 |
| | | Summer | 1.82 | 0.02 | 2.15 | 0.00 | 2.09 | -0.02 | 2.15 | 0.07 | 1.14 | 0.04 |
| | | Fall | 2.33 | 0.16 | 2.05 | 0.12 | 1.76 | 0.11 | 2.29 | 0.20 | 0.03 | 1.72 |
| | | Winter | 1.92 | 0.11 | 1.99 | 0.02 | 1.84 | 0.31 | 1.80 | 0.02 | 1.88 | 0.18 |
| | by site Roadside | MK | **1.96** | **-1.24** | 2.00 | -1.29 | 1.78 | -0.91 | 1.51 | -1.02 | 1.83 | -1.11 |
| | Urban | TW | **2.02** | **-0.12** | 2.01 | -0.13 | 1.94 | -0.13 | 2.15 | -0.14 | 2.10 | -0.10 |
| | | YL | **2.14** | **-0.13** | 2.01 | -0.05 | 2.08 | -0.11 | 1.90 | 0.03 | 2.33 | -0.20 |
| | | CW | **2.42** | **-0.14** | 2.51 | -0.19 | 2.4 | 0.16 | 2.73 | -0.23 | 2.31 | -0.10 |
| | | TC | **2.26** | **-0.01** | 2.23 | 0.00 | 2.25 | -0.03 | 1.98 | 0.08 | 2.35 | -0.02 |
| | | Urban sites combined | **2.11** | **-0.03** | 2.10 | -0.03 | 2.09 | -0.04 | 2.09 | 0.00 | 2.13 | -0.03 |
| | Suburban | WB | **2.65** | **0.11** | 2.74 | 0.07 | 2.86 | 0.03 | 2.47 | 0.14 | 2.70 | 0.14 |

[a]**The two reconstruction method equations are:**

**M1**: $\qquad EC_{IMP\_TOR} = a \times EC_{NSH\_TOT} + b$

**M2**: $\qquad EC_{IMP\_TOR} = AEC_{NSH} + OC4_{NSH} - (a \times PC_{NSH\_TOR} + b)$

[b] Regression parameters are derived from 2011-2013 data.

[c] Regression parameters are derived from 2011-2012 data.

[d] Regression parameters are derived from a single year's data.




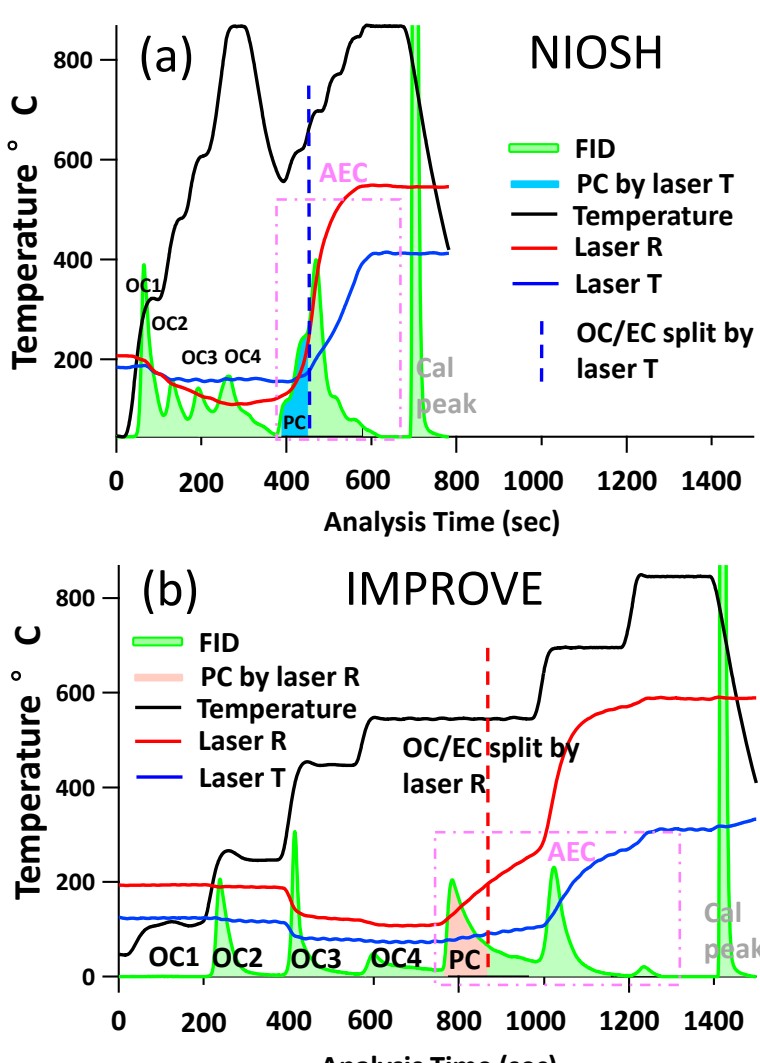


Figure 1 Thermograph of typical thermal optical analysis (Sample CW20130118) using a Sunset carbon

analyzer. (a) NIOSH protocol (b) IMPROVE protocol (FID: flame ionization detector signal; PC: pyrolysis

carbon; AEC: apparent EC, which is the sum of all the EC fractions before correcting for PC; Temperature:

Oven temperature during analysis; Laser T: laser transmittance signal; Laser R: laser reflectance signal;

Cal peak: calibration peak at the end of each analysis)






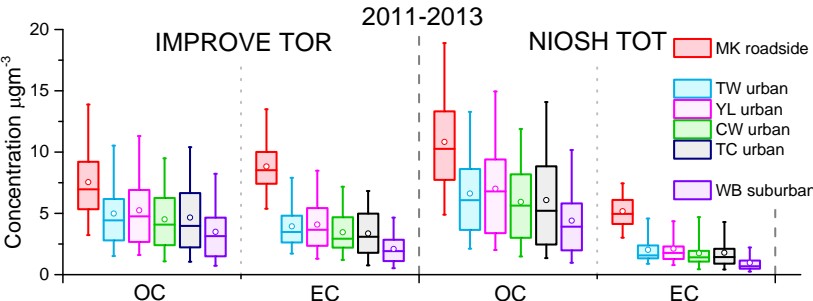


Figure 2 Three-year distributions of OC and EC concentrations by IMPROVE TOR and NIOSH TOT

protocols. The symbols in the boxplots represent the average (open circles), median (interior lines), 75th and the 25th percentile (box boundaries), and 95th and 5th percentile (whiskers).





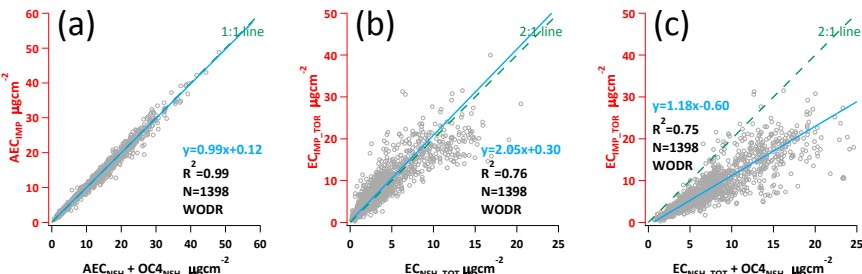


Figure 3 Comparison of different carbon fractions. (a) Relationship of IMPROVE apparent EC (AEC$_{IMP}$,

sum of EC1$_{IMP}$ to EC3$_{IMP}$) and the sum of NIOSH apparent EC (AEC$_{NSH}$, sum of EC1$_{NSH}$ to EC6$_{NSH}$) plus OC4$_{NSH}$. (b) Relationship of EC$_{IMP\_TOR}$ (y axis) and EC$_{NSH\_TOT}$ (x axis) (c) Relationship of EC$_{IMP\_TOR}$ (y
axis) and the sum of EC$_{NSH\_TOT}$ and OC4$_{NSH}$ (x axis)






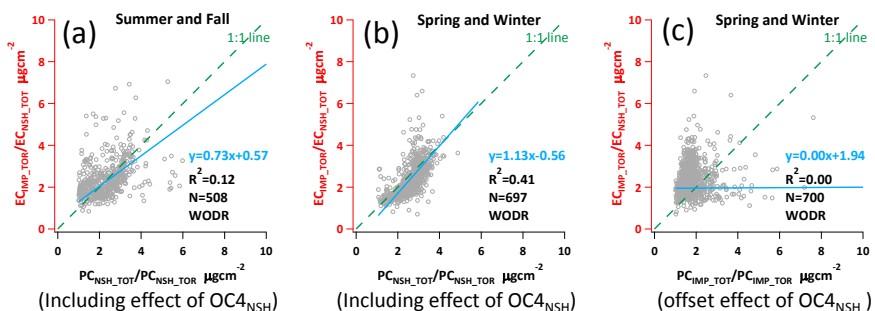

Figure 4 $EC_{IMP\_TOR}$ to $EC_{NSH\_TOT}$ discrepancy dependency on TOT/TOR charring correction. (a) $EC_{IMP\_TOR}$ /$EC_{NSH\_TOT}$ vs. $PC_{NSH\_TOT}$/$PC_{NSH\_TOR}$ ratio-ratio plot for summer and fall (b) $EC_{IMP\_TOR}$
/$EC_{NSH\_TOT}$ vs. $PC_{NSH\_TOT}$/$PC_{NSH\_TOR}$ ratio-ratio plot for spring and winter (c) $EC_{IMP\_TOR}$ /$EC_{NSH\_TOT}$ vs. $PC_{IMP\_TOT}$/$PC_{IMP\_TOR}$ ratio-ratio plot for spring and winter





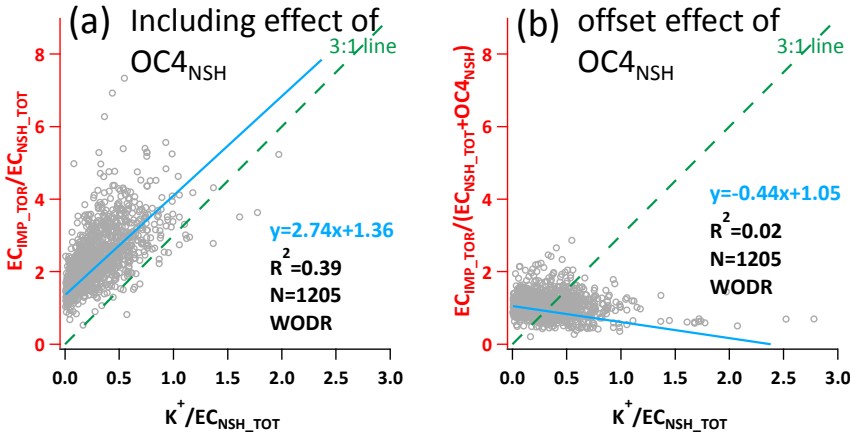


Figure 5 $EC_{IMP\_TOR}$ to $EC_{NSH\_TOT}$ discrepancy dependency on $K^+/EC_{NSH\_TOT}$ ratio. (a) $EC_{IMP\_TOR}$
$/EC_{NSH\_TOT}$ vs. $K^+/EC_{NSH\_TOT}$ ratio-ratio plot (b) $EC_{IMP\_TOR}/(EC_{NSH\_TOT}+OC4_{NSH})$ vs. $K^+/EC_{NSH\_TOT}$
        ratio-ratio plot




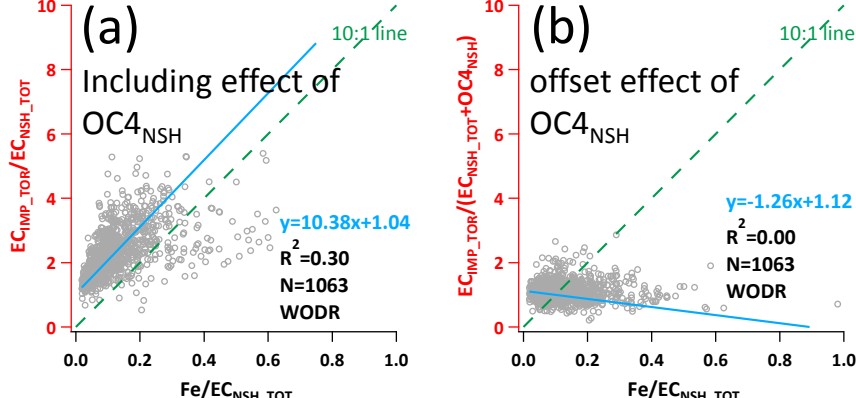

Figure 6 $EC_{IMP\_TOR}$ to $EC_{NSH\_TOT}$ discrepancy dependency on $Fe/EC_{NSH\_TOT}$ ratio. (a) $EC_{IMP\_TOR}$ /$EC_{NSH\_TOT}$ vs. $Fe/EC_{NSH\_TOT}$ ratio-ratio plot (b) $EC_{IMP\_TOR}/(EC_{NSH\_TOT}+OC4_{NSH})$ vs. $Fe/EC_{NSH\_TOT}$
ratio-ratio plot





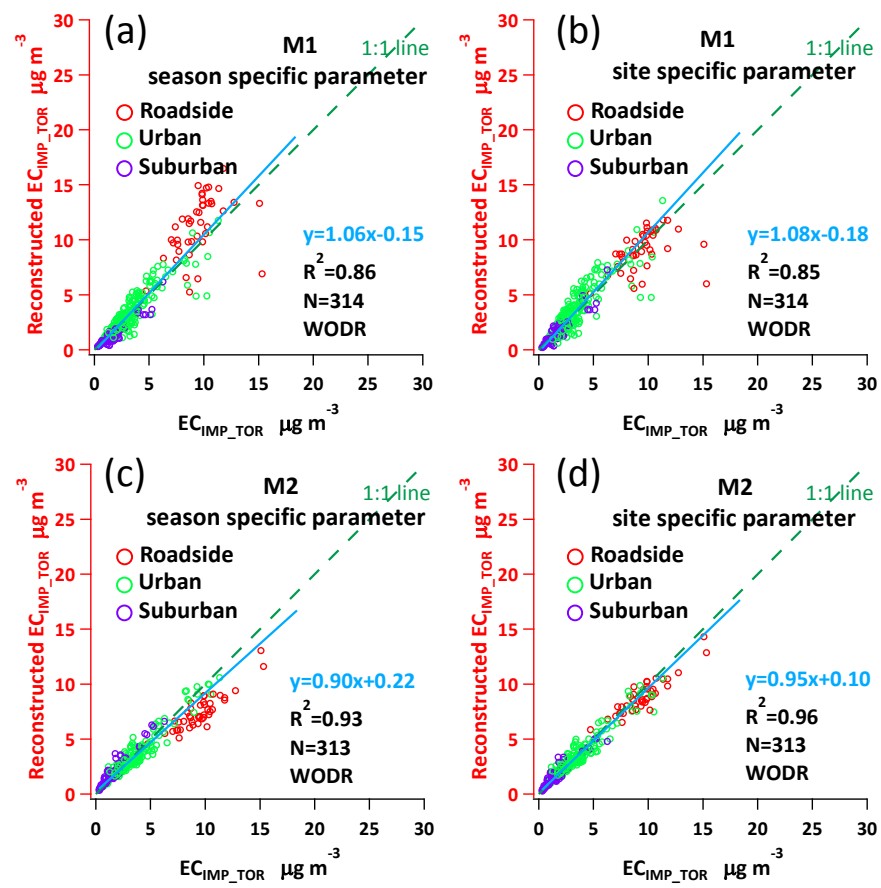

Figure 7 Comparison of reconstructed $EC_{IMP\_TOR}$ and measurement $EC_{IMP\_TOR}$ in year 2013. (a) Regression by season-specific parameters using M1. (b) Regression by site-specific parameters using M1. (c) Regression by season-specific parameters using M2. (d) Regression by site-specific parameters using M2.







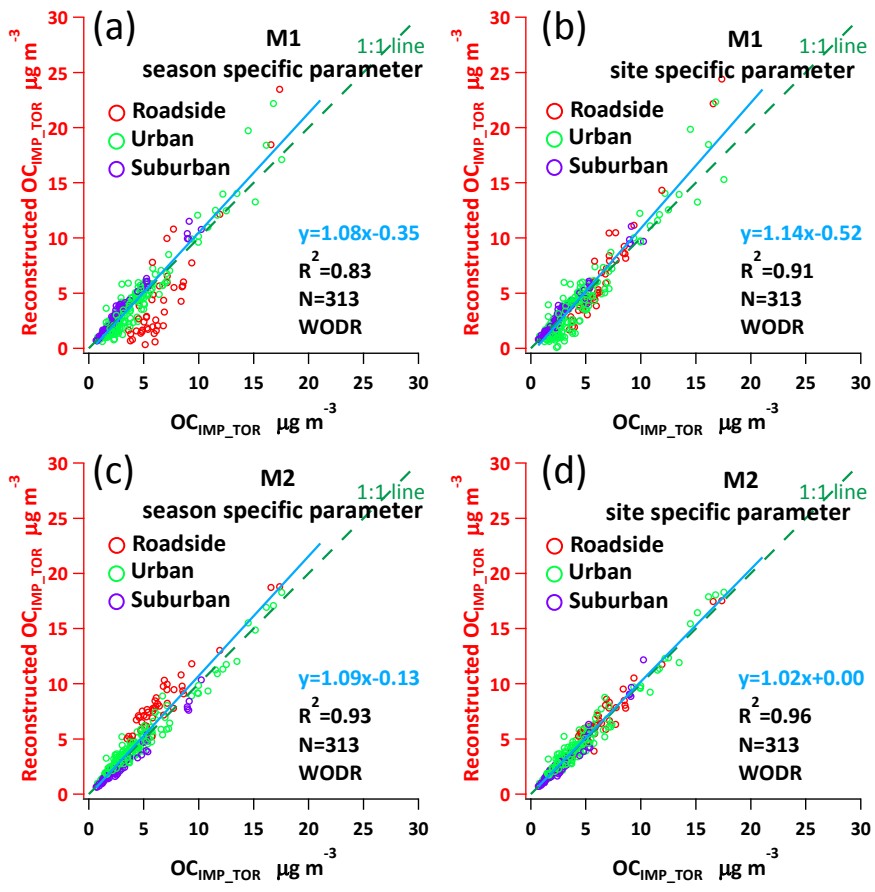

Figure 8 Reconstruction of $OC_{IMP\_TOR}$ calculated using Equation 8. (a) Reconstruction by season-specific
        parameters using M1. (b) Reconstruction by site-specific parameters using M1. (c) Reconstruction by
season-specific parameters using M2.   (d) Reconstruction by site-specific parameters using M2.





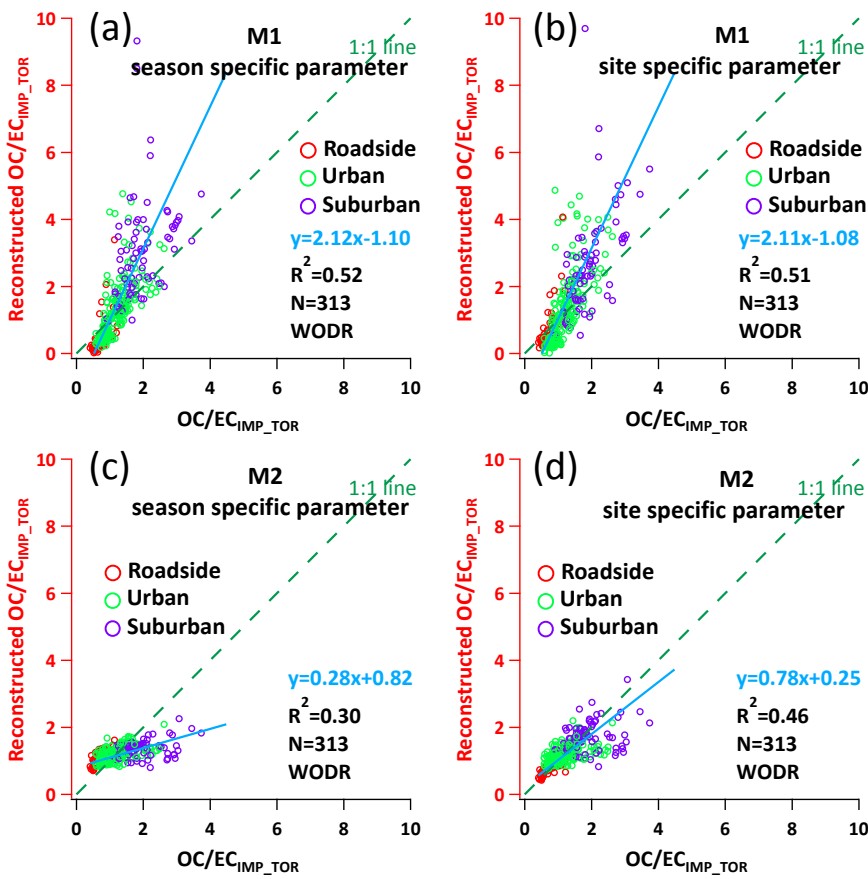

Figure 9 Reconstruction results of OC/EC_{IMP\_TOR}. (a) Reconstruction by season-specific parameters using
       M1. (b) Reconstruction by site-specific parameters using M1. (c) Reconstruction by season-specific
parameters using M2. (d) Reconstruction by site-specific parameters using M2.






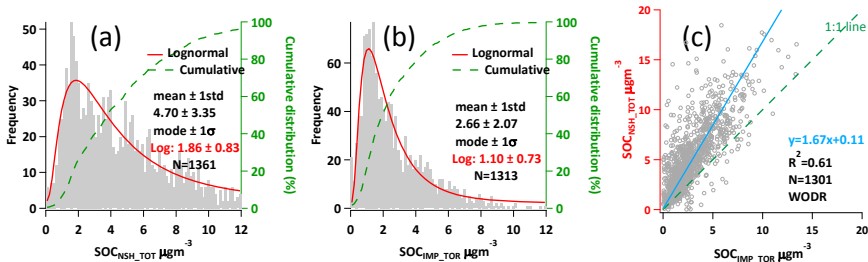

Figure 10 Comparison of SOC by NIOSH and IMPROVE. (a) SOC estimation from NIOSH TOT data. (b)
SOC estimation from IMPROVE TOR data. (c) Relationship between $SOC_{NSH\_TOT}$ and $SOC_{IMP\_TOR}$.






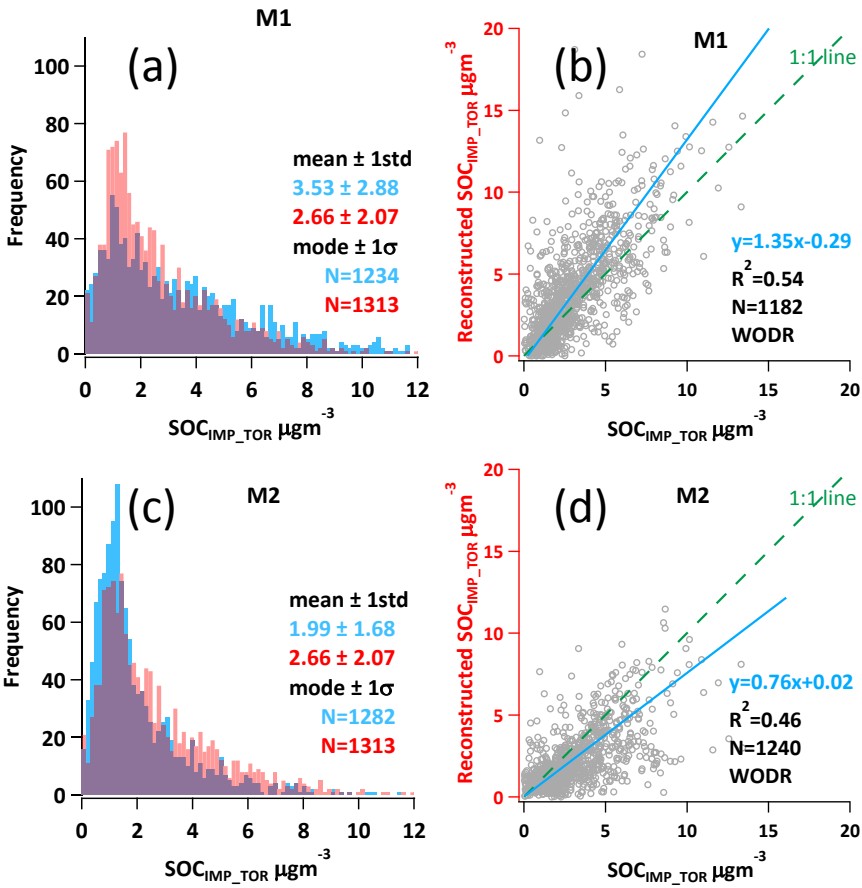


Figure 11. Histogram comparison of original SOC$_{IMP\_TOR}$ (in red) with reconstructed SOC$_{IMP\_TOR}$ (in blue):

(a) by M1. (c) by M2    Scatter plot comparison of original SOC$_{IMP\_TOR}$ (in x axis) with reconstructed SOC$_{IMP\_TOR}$ (in y axis): (b) by M1, (d) by M2