# Peer review of "Inter-comparison of NIOSH and IMPROVE protocols for OC and EC determination: Implications for inter-protocol data conversion"

_Atmospheric Measurement Techniques, 2016_

## Referee Comment (RC1) · Anonymous Referee #2 · 14 Jun 2016

The paper describes an inter-comparison of NIOSH and IMPROVE temperature protocols and potentials on data conversion between the two. Even though the topic has been the center of several other works it is certainly an important addition when considering the quantity of samples included and the further investigation of the applicability of conversion equations, including other parameters like biomass burning tracers, SOC, potassium and ferric oxides. Overall an orderly and thorough work that deepens in the comparability of the two protocols; therefore, I suggest the acceptance of the paper for publication after dealing with the following comments and remarks:

2.2 Sample analysis: Temperature offsets of up to 100 oC have been observed to be present and vary per instrument, protocol and temperature step. Was the procedure

of temperature offsets' correction considered in this study? Were there one or several ovens and heating coils installed? Oven soiling and aging has been also found to have an influence on results. Related studies for consideration:

Chiappini et al.: Clues for a standardized thermal-optical protocol for the assessment of organic and elemental carbon within ambient air particulate matter, Atmos. Meas. Tech., 7, 1649–1661, doi:10.5194/amt-7-1649-2014, 2014.

Panteliadis et al.: ECOC comparison exercise with identical thermal protocols after temperature offset correction – instrument diagnostics by in-depth evaluation of operational parameters, Atmos. Meas. Tech., 8, 779–792, doi:10.5194/amt-8-779-2015, 2015.

Pavlovic, J., Kinsey, J. S., and Hays, M. D.: The influence of temperature calibration on the OC–EC results from a dual-optics thermal carbon analyzer, Atmos. Meas. Tech., 7, 2829-2838, doi:10.5194/amt-7-2829-2014, 2014.

3.2 NIOSH and IMPROVE comparison: OC4 ramps up from 615 oC to 870 oC. This means that the fraction evolving from 550 oC to 615 oC is not included in the equation. Doesn't this introduce an error to the equation 1?

Further, in line 173, the term "laser effect" is introduced, which refers directly to the actual instrument part/laser unit and not to the optical method selected which is implied here. Please consider replacing with "optical method effect", at this point and also further down the text. I am not convinced though that the thermal and optical effects are independent and can be separated. Isn't the PC that is responsible for the so called "laser effect" generated thermally, earlier, during the inert phase?

Lines 190-200: This paragraph seems a bit difficult for the reader to follow. Could you please rephrase?

Finally, from this chapter and Figure 3 I assume that PC-IMP is higher from PC-NSH by ∼18%. Could you discuss a bit more why is this observed?

Line 134: Your criteria seem to be very loose, especially when considering the distribution of results on Figure S2b. What is the reasoning behind this? Could these criteria be stricter?

3.3.1: I would prefer M2-1 reconstruction method included in the main text. According to my opinion it points out the importance of monitoring other relevant factors like Fe and K+ next to OCEC analysis and further shows a good fitting, certainly adding value to the paper.

Line 53: This sentence could be improved. Maybe continuous monitoring of laser transmittance?

Line 56: What is meant by "sampling" networks? Monitoring?

Line 101: Were the filters collected every 24 hours manually or was there a sequential sampler installed? Not clear.

Table1: The "Overall" column would be better as the last column. Does overall refer to average? Please indicate also which measurements are made with NIOSH and which with Improve in the table. Further, some measurements were performed with a different analyzer. Maybe worthwhile inserting a footnote with this information?

Correct NISOH to NIOSH. Couple occasions in the text.

Line 52: Pure He reads better than He-only

Line 58: Protocols also differ in duration.

Line 102: Remove "."

Line 144: $\mu$gm-3 to $\mu$g m-3

Line 257: The abbreviation RHS is not introduced.

Line 348: Both to both

Line 358: 2.66$\mu$g to 2.66 $\mu$g

Table 2: Some figures are in bold. Is there a reason for this?

Figure 1: a) The OC/EC split line in the legend is vertical while the rest horizontal b) The line of the OC/EC split is not included in the legend

Table S3: Does "Measured OC and EC" mean OC/EC?

Figure S4: Is PC included in one of the fractions mentioned here?

The supplemental material seems too extensive. Would you consider removing some figures? eg S16 to S20.

---

## Referee Comment (RC2) · Anonymous Referee #1 · 5 Jul 2016

This paper aims to assess the disagreement on EC results between the IMPROVE and the NIOSH thermal protocol and investigates the reasons for this discrepancy. The authors suggest various reconstruction methods to perform OC and EC inter-protocol data conversion with the purpose to further exploit the current OC and EC datasets. This work is certainly relevant to the scope of AMT and the methods presented are sound and in general well described. Although the manuscript is well structured some sections (section 3.2 and 3.3.2) are difficult to follow as too many figures are given both in the main manuscript and in the supplementary material. I recommend it for publication in AMT after the comments below are addressed.

Comments

[Figure]

Introduction: The authors should mention that the there is no unique NIOSH protocol since many NIOSH-type protocols exist, with maximum temperatures in the inert mode found in the range 820‐900 °C.

Section 2.3 Please provide the exact number of valid data

Line 142: Also the residence time is different as the IMPROVE protocol advances from one temperature to the next one when a well-defined carbon peak has evolved

Line 154: This is not true: In the NIOSH protocol the carbon mass evolving from 550 °C to 870 °C represents part of the OC3 peak and the OC4 peak. Equation (1) should be corrected to include the manually integrated area from 550 to 870 °C and not only the OC4 peak. The authors should explain why they have included only the OC4 peak in the equation.

Line 172: What do you mean by thermal effect and laser effect? Both OC4 peak and PC formation depends on the aerosol chemical composition and the temperature steps and residence time in the inert mode. Correction for charring is achieved by monitoring of transmittance or reflectance.

Lines 201-226: I find this paragraph a bit difficult to follow. Could you please simplify it?

Line 255: What does RHS represent?

Section 3.3: As one of the objectives of this paper is to estimate the EC IMP_TOR from NIOSH TOT data it would better to include all reconstruction methods in the main manuscript

Conclusions: The authors should mention somewhere in the text that all NIOSH TOT analysis should have been done by the same analyzer otherwise other instrument specific parameters might influence the regression.

Table 1: Please include "mean" in the caption

---

## Author Comment (AC1) · 16 Aug 2016

see the attached file.

Please also note the supplement to this comment:
http://www.atmos-meas-tech-discuss.net/amt-2016-116/amt-2016-116-AC1-supplement.pdf

---

## Author Comment (AC2) · 16 Aug 2016

**Point-by-point response to review comments on manuscript amt-2016-116 "Inter-comparison of NIOSH and IMPROVE protocols for OC and EC determination: Implications for inter-protocol data conversion"**

**By Cheng Wu et al.**

**Anonymous Referee #2**

The paper describes an inter-comparison of NIOSH and IMPROVE temperature protocols and potentials on data conversion between the two. Even though the topic has been the center of several other works it is certainly an important addition when considering the quantity of samples included and the further investigation of the applicability of conversion equations, including other parameters like biomass burning tracers, SOC, potassium and ferric oxides. Overall an orderly and thorough work that deepens in the comparability of the two protocols; therefore, I suggest the acceptance of the paper for publication after dealing with the following comments and remarks:

**Author's Response:** We thank the reviewer for the positive assessment of our work.

2.2 Sample analysis: Temperature offsets of up to 100 °C have been observed to be present and vary per instrument, protocol and temperature step. Was the procedure of temperature offsets' correction considered in this study? Were there one or several ovens and heating coils installed? Oven soiling and aging has been also found to have an influence on results. Related studies for consideration:

Chiappini et al.: Clues for a standardized thermal-optical protocol for the assessment of organic and elemental carbon within ambient air particulate matter, Atmos. Meas. Tech., 7, 1649–1661, doi:10.5194/amt-7-1649-2014, 2014.

Panteliadis et al.: ECOC comparison exercise with identical thermal protocols after temperature offset correction – instrument diagnostics by in-depth evaluation of operational parameters, Atmos. Meas. Tech., 8, 779–792, doi:10.5194/amt-8-779-2015, 2015.

Pavlovic, J., Kinsey, J. S., and Hays, M. D.: The influence of temperature calibration on the OC–EC results from a dual-optics thermal carbon analyzer, Atmos. Meas. Tech., 7, 2829-2838, doi:10.5194/amt-7-2829-2014, 2014.

**Author's Response:** All the samples are analyzed on the same Sunset carbon analyzer for both NIOSH and IMPROVE protocols. In the course of the three years' ECOC analysis, three ovens were used. No heating coil was replaced. After each oven replacement, the thermocouple of the front oven was calibrated at a fixed temperature and the offset was adjusted accordingly. Instrument calibrations were routinely carried out on a biweekly basis and always after any maintenance work, such as replacing an oven or heating coils; however, we did not monitor whether oven soiling and aging impacts the ECOC split. We did not conduct calibration of temperature of the sample boat on the Sunset analyzer, as the sample boat temperature might be different from the temperature sensor for the oven. We agree that potential temperature offset would be a source of uncertainties for the inter-protocol

comparison. The following text is added to clarify the point raised here and to make note of the limitations.

Lines 117-118:

"All the OCEC samples were analyzed on the same Sunset analyzer using both NIOSH and IMPROVE protocols."

Lines 445-449:

"It should be noted that the conversion equations established in this work are based on that all the ECOC data analysis are done by the same analyzer. Other instrument specific parameters might influence the regression if multiple instruments are used in obtaining the OCEC data. For example, temperature offset has been found varied by instrument in different labs (Panteliadis et al., 2015). Oven soiling and aging has also been found to have an optical influence that introduces uncertainties to the results."

3.2 NIOSH and IMPROVE comparison: OC4 ramps up from 615 °C to 870 °C. This means that the fraction evolving from 550 °C to 615 °C is not included in the equation. Doesn't this introduce an error to the equation 1?

**Author's Response:** Reviewer #1 made the same comment. We copy below our response to this comment.

We note that both temperature and duration of each temperature step affect the amount of OC evolved corresponding to each temperature steps (i.e., OC1, OC2, OC3, and OC4). Due to the much longer step durations in the IMPROVE protocol, carbon evolved beyond $550^{o}C$ in IMPROVE protocol does not simply map to OC evolved beyond $550^{o}C$ in NIOSH protocol, which includes part of OC3 and OC4). The equivalence of IMPROVE AEC and the sum of NIOSH OC4 and NIOSH AEC, as indicated by Eq. (1), is established through comparing actual data in current study (Fig. 3a showing a slope of 0.99) and a previous study of ours (Wu et al., 2012). The near unity slope seen in Fig. 3a indicates that the temperature point for establishing the equivalence was not $550^{o}C$, instead, it corresponds to the last temperature step of the protocol (i.e., $870^{o}C$). The relevant text (lines 160-163) is re-worded and new text (lines 168-171) is added to improve the clarity.

Further, in line 173, the term "laser effect" is introduced, which refers directly to the actual instrument part/laser unit and not to the optical method selected which is implied here. Please consider replacing with "optical method effect", at this point and also further down the text. I am not convinced though that the thermal and optical effects are independent and can be separated. Isn't the PC that is responsible for the so called "laser effect" generated thermally, earlier, during the inert phase?

**Author's Response:** We have adopted the suggestion to replace "laser effect" with "optical method effect" in the revised manuscript. Text is added to clearly explain what is thermal effect and what is optical method effect (lines 184-186). We agree that thermal and optical

effects are not independent. While a clean cut between the two effects did not exist, we think the attribution of thermal and optical effects demonstrated here still can serve as an apparent indicator for the relative contributions from thermal and optical effects.

Lines 184-186:
"Thermal effect refers to inter-protocol EC difference caused by temperature steps difference. Optical method effect is inter-protocol EC difference introduced by PC difference between transmittance and reflectance charring correction."

Lines 190-200: This paragraph seems a bit difficult for the reader to follow. Could you please rephrase?

**Author's Response:** We rephrased the following sentences in this paragraph.

Lines 210-213:
"This selective dependency suggests the inter-protocol PC difference introduced by TOT/TOR is more pronounced when PC contains char formed in $OC4_{NSH}$ stage. Since $PC_{NSH}$ contains char formed in the $OC4_{NSH}$ stage while $PC_{IMP}$ is not, $OC4_{NSH}$ is the major difference between potential sources of $PC_{IMP}$ and $PC_{NSH}$ difference."

Finally, from this chapter and Figure 3 I assume that PC-IMP is higher from PC-NSH by ~18%. Could you discuss a bit more why is this observed?

**Author's Response:** The slope 1.18 in Figure 3c implies that $EC_{IMP\_TOR}$ is higher than $EC_{NSH\_TOT}+OC4_{NSH}$ by 18%. Incorporating Eq.3 and Eq. 5, we can see that $PC_{NSH\_TOT}$ is higher than $PC_{IMP\_TOR}$. Two factors are responsible for this result. Firstly, NIOSH protocol has higher peak inert mode temperature (870 vs 550 °C) than IMPROVE, resulting more char formation by NIOSH protocol. Secondly, laser reflectance signal returns to initial level earlier than the transmittance signal. This is because the reflectance signal only accounts for the char at the surface of filter, which evolves first when oxygen is introduced. In comparison, the transmittance signal accounts for char throughout the filter, making transmittance signal's return relatively later than reflectance.

Line 134: Your criteria seem to be very loose, especially when considering the distribution of results on Figure S2b. What is the reasoning behind this? Could these criteria be stricter?

**Author's Response:** We conduct a test by tightening the criteria from $0.5<TC_{NSH}/TC_{IMP}<2$ to $0.8<TC_{NSH}/TC_{IMP}<1.2$, the valid data points decrease from 1398 to 1377 (1.5% reduction). The improvement in TC (Figure S2a) is almost negligible (as shown in Figure R1). Comparison of data screening on $EC_{IMP\_TOR}$ reconstruction using M1 are shown in Figure R2, the results are almost identical. As a result, we feel that the existing criteria can satisfy the comparison purpose and there is no need to make it stricter.

[Figure]

Figure R1 Comparison of total carbon quantification by IMPROVE TOR and NIOSH TOT protocols. (a) 0.5<$TC_{NSH}$/$TC_{IMP}$<2 (b) 0.8<$TC_{NSH}$/$TC_{IMP}$<1.2

[Figure]

Figure R2 Reconstruction of $OC_{IMP\_TOR}$ calculated using Equation 7. (a) 0.5<$TC_{NSH}$/$TC_{IMP}$<2 (b) 0.8<$TC_{NSH}$/$TC_{IMP}$<1.2

3.3.1: I would prefer M2-1 reconstruction method included in the main text. According to my opinion it points out the importance of monitoring other relevant factors like Fe comment and K+ next to OCEC analysis and further shows a good fitting, certainly adding value to the paper.

**Author's Response:** We have taken the suggestion to include the description of M2-1 in the main text.

Line 53: This sentence could be improved. Maybe continuous monitoring of laser transmittance?

**Author's Response:** Suggestion taken. The said sentence is changed to following:

Line 53-54:
"A correction for charred OC (pyrolysis carbon, PC) in the inert stage relies on continuous monitoring of laser transmittance or reflectance of the filter."

Line 56: What is meant by "sampling" networks? Monitoring?

**Author's Response:** "sampling" was changed to "monitoring".

Line 101: Were the filters collected every 24 hours manually or was there a sequential sampler installed? Not clear.

**Author's Response:** Each sampling event was programmed one day before and the sample collection duration is midnight to midnight. The filters were then retrieved back to the lab the next day. Although a sequential sampler was installed at each site, the sequential sampling function was not used. The relevant text is revised to be clearer about the sampling details.

Lines 98-99:
"One 24-hour $PM_{2.5}$ sample (from midnight to midnight) was programmed and collected every six days from January 2011 to December 2013…"

Table1: The "Overall" column would be better as the last column. Does overall refer to average?
Please indicate also which measurements are made with NIOSH and which with Improve in the table. Further, some measurements were performed with a different analyzer. Maybe worthwhile inserting a footnote with this information?

**Author's Response:** "Overall" is referring 3-year average. Table was revised accordingly.

Correct NISOH to NIOSH. Couple occasions in the text.

**Author's Response:** Typo corrected.

Line 52: Pure He reads better than He-only

**Author's Response:** Suggestion taken (Lines 30, 52).

Line 58: Protocols also differ in duration.

**Author's Response:** The sentence is changed as follows.

Line 58:
"…which differ in their temperature ramping, steps duration and optical correction schemes (Table S1)."

Line 102: Remove "." Line 144: µgm-3 to µg m-3

**Author's Response:** Revisions made.

Line 257: The abbreviation RHS is not introduced.

**Author's Response:** "RHS" was changed to "right hand side"

Line 348: Both to both

**Author's Response:** Revisions made.

Line 358: 2.66µg to 2.66 µg

**Author's Response:** Revisions made.

Table 2: Some figures are in bold. Is there a reason for this?

**Author's Response:** It is used to indicate the recommended parameters for future applications in Hong Kong. We added a recommendation section instead.

Figure 1: a) The OC/EC split line in the legend is vertical while the rest horizontal b) The line of the OC/EC split is not included in the legend

**Author's Response:** Revisions made.

Table S3: Does "Measured OC and EC" mean OC/EC?

**Author's Response:** Sorry for the confusion. "Measured OC and EC" was changed to "$(OC/EC)_{pri}$ calculated from measured OC and EC". "reconstructed OC and EC" was changed to "$(OC/EC)_{pri}$ calculated from reconstructed OC and EC"

Figure S4: Is PC included in one of the fractions mentioned here?

**Author's Response:** Yes, PC is usually part of EC1 and EC2 as illustrated in Figure 1.

The supplemental material seems too extensive. Would you consider removing some figures? eg S16 to S20.

**Author's Response:** We prefer to keep these figures (Figs. S16-S20) as they provide visual comprehension of comparisons between measured and reconstructed OC, EC and OC/EC data.